# 3D Road Boundary Extraction Based on Machine Learning Strategy Using LiDAR and Image-Derived MMS Point Clouds

**DOI:** 10.3390/s24020503

**Published:** 2024-01-13

**Authors:** Baris Suleymanoglu, Metin Soycan, Charles Toth

**Affiliations:** Department of Civil, Environmental and Geodetic Engineering, The Ohio State University, 470 Hitchcock Hall, 2070 Neil Ave., Columbus, OH 43210, USA; bariss@yildiz.edu.tr (B.S.); soycan@yildiz.edu.tr (M.S.)

**Keywords:** mobile mapping systems, mobile laser scanning, curb detection, 3D road extraction, machine learning

## Abstract

The precise extraction of road boundaries is an essential task to obtain road infrastructure data that can support various applications, such as maintenance, autonomous driving, vehicle navigation, and the generation of high-definition maps (HD map). Despite promising outcomes in prior studies, challenges persist in road extraction, particularly in discerning diverse road types. The proposed methodology integrates state-of-the-art techniques like DBSCAN and RANSAC, aiming to establish a universally applicable approach for diverse mobile mapping systems. This effort represents a pioneering step in extracting road information from image-based point cloud data. To assess the efficacy of the proposed method, we conducted experiments using a large-scale dataset acquired by two mobile mapping systems on the Yıldız Technical University campus; one system was configured as a mobile LiDAR system (MLS), while the other was equipped with cameras to operate as a photogrammetry-based mobile mapping system (MMS). Using manually measured reference road boundary data, we evaluated the completeness, correctness, and quality parameters of the road extraction performance of our proposed method based on two datasets. The completeness rates were 93.2% and 84.5%, while the correctness rates were 98.6% and 93.6%, respectively. The overall quality of the road curb extraction was 93.9% and 84.5% for the two datasets. Our proposed algorithm is capable of accurately extracting straight or curved road boundaries and curbs from complex point cloud data that includes vehicles, pedestrians, and other obstacles in urban environment. Furthermore, our experiments demonstrate that the algorithm can be applied to point cloud data acquired from different systems, such as MLS and MMS, with varying spatial resolutions and accuracy levels.

## 1. Introduction

Roads form an important part of the transportation network, and their geometric model is essential for many applications, such as driver assistance and safety warning systems [1], high-precision maps, and infrastructure planning and maintenance [2]. Accurate and regularly updated road information, i.e., geometry, topography, and the classification of the road corridor, implemented in CAD/GIS systems is required to support road inspection, road asset inventory, road safety analysis, and more recently autonomous driving [3] and achieving city modeling and/or urban planning [4].

The extraction of road information from remote-sensed data has been one of the most important research topics in the transportation infrastructure field in recent years. Several sensors and measurement techniques have been used to extract road information, including conventional survey techniques (GPS/GNSS, total station), Airborne Laser Scanning (ALS), satellite and aerial imagery, mobile mapping, etc. [5]. The classical measurement techniques, such as using GPS/GNSS and total station, are extremely labor-intensive and, most importantly, can be dangerous in areas with heavy traffic [6]. Using satellite or aerial imagery is safe and productive, although it could be negatively affected by weather conditions and offers somewhat modest spatial resolution, and thus, limiting the detection objects of small spatial sizes, such as utility poles, curb lines, traffic signs, and lamp posts [7].

The light detection and ranging (LiDAR) is a powerful remote sensing technique that captures detailed 3D geometrical as well as some radiometric information of the objects in a scene [8]. The point density of the LiDAR-acquired point cloud is defined in angular resolution, and thus, the density on object surfaces declines with the object distance at the rate of 1/range^2^. Therefore, from the various LiDAR platforms, mobile laser scanning systems (MLS), both vehicle and UAS platforms, are considered the primary technology for acquiring adequate 3D point cloud data to support road-related information extraction, see relevant studies [9,10,11].

Recent advancements in positioning and imaging sensors, computer software and hardware, and particularly, image-based 3D reconstruction algorithms have sparked growing interest in the development of low-cost MMS based on photogrammetric techniques [12,13]. In addition, the decreasing costs of imaging sensors has made photogrammetry-based MMS systems an extremely attractive alternative to LiDAR-based systems in terms of cost and easier-to-operate logistics [11].

Although numerous studies have investigated road extraction and detection techniques, extracting roads from mobile mapping data remains a challenging task due to the complex and diverse nature of road environments. Achieving good performance generally requires using a combination of different techniques and a careful tuning of algorithm parameters. More recently, the integration of artificial intelligence technologies has offered new ways to improve the performance of existing methods [14]. In this study, we aim to develop a novel methodology based on machine learning techniques to detect and extract 3D road boundaries obtained from both MLS- and MMS-created point clouds. To evaluate the effectiveness of the proposed methodology, we compared the results obtained for both systems against manually measured reference road boundary information, including a detailed cross performance analysis of the advantages and disadvantages of each system. This paper is organized as follows: Section 2 presents the related studies on the topic, Section 3 describes the proposed methodology, and Section 4 provides a detailed evaluation of the experimental results. Finally, in Section 5, we present our conclusions and discuss the potential future research.

The main contribution of this paper are as follows:We propose a new methodology that integrates machine learning for the extraction of road boundaries and surfaces from point clouds, utilizing object space constraints applicable to both MLS and MMS data;We present a novel approach for road edge segmentation that enables the detection of various road structures, such as curbs, turns at intersections, traffic islands, and roundabouts;We conducted experiments with dataset acquired from a MLS and a MMS and assessed the accuracy of the results by comparing with reference ground truth, manually measured with the RTK-GNSS technique.

## 2. Related Studies

Mi et al. [15] classified road boundary extraction methods into three categories, such as activity-driven, feature-driven, and model-driven, although methods are also grouped based sensor data, such as LiDAR or optical imagery. Activity-driven methods take advantage of vehicle motion (trajectory information) to classify road and non-road regions [16]. Feature-driven methods use the distribution of road surface properties, such as surface smoothness, flatness, etc., or local patterns, such as curb height, to detect road boundaries. These methods are also called 3D-based methods by some authors [3,5]. Model-driven methodologies extract road boundaries based on geometric models, as foundational information. Utilizing this prior knowledge, they estimate road boundary information through diverse parametric models such as clothoid and ribbon snake. In the final step, based on the generated boundary estimations [17,18], the road data are grouped into road and non-road segments.

Activity-driven methods use the mobile sensor data generated by vehicles to determine road boundaries [19,20]. These algorithms categorize road segments by analyzing GPS/GNSS trajectories [21]. However, the road boundaries determined in this way may not be sufficient to achieve the required accuracy and reliability standards [15].

Several feature-driven methods were developed to extract road boundaries and surfaces. Manandhar and Shibasaki [22] developed an algorithm to extract road surfaces from MLS point clouds based on elevation differences, point density, and slope change. Ibrahim and Lichti [23] determined curb and street floor depending on the change in point density. In addition, different MLS data features, such as intensity, laser pulse waveform, height and slope difference, and point patterns, have been used to detect and extract the road surface [24,25,26,27]. Similarly, Yang et al. [28] divided the point cloud into multiple cross-sections using GPS time. Then, using a moving window operator, the road boundaries were determined based on the height difference, slope change, and point density information. Similarly, Guan et al. [29] used the trajectory data to segment point cloud data into blocks, and then, curb points are extracted using both slope and elevation differences. Xu et al. [30] developed a two-step method to detect curb points. In this context, the curb candidate points were detected using the density gradient-based energy function, and the road boundaries are refined using the least-cost path model. Zai et al. [5] developed a supervoxel-based method to detect curb points with graph cut algorithm.

Model-based methods primarily focus on extracting road boundaries through model adaption, which makes the use of pre-existing information, such as road designs, dimensions, and curb heights. Initially, a parametric representation, such as a linear model [31,32], a clothoid model [17], or a ribbon snake model [18], is used to identify the shape of the road border. Next, a comprehensive boundary forecast based on the previous boundary model is created with different point labels, indicating roads or off-road areas. In this regard, Huang et al. [33] examined general road shapes, including the central axis and a typical platform width, for curb extraction and tracing. However, fixed road width and central axis constraints may not accurately represent road morphology, particularly at intersections and road segments with varying lane numbers.

Mobile mapping systems have experienced major developments in the past decade. While mobile LiDAR-based systems (MLS) dominated the industry, low-cost image-based mobile mapping systems (MMS) have gained popularity recently [34]. These systems employ various application-specific camera sensors and configurations such as monocular cameras, RGB-D cameras, or multi-camera systems. Monocular cameras generate a sequence of RGB images without directly providing depth information [35]. For instance, Aufrere et al. [36] utilized histograms derived from distance measurements acquired with a camera and a laser line stripper vision system. Yamaguchi et al. [37] developed a method to analyze the road area from images taken with a vehicle-mounted monocular camera. Unfortunately, these methods perform poorly in suboptimal conditions and produce a large number of false positive results [38,39]. Additionally, these approaches may fall short of generating a 3D scale or establishing 3D points [14]. For road pavement detection and assessment, stereo vision systems are frequently used to obtain depth estimates from images through disparity maps. These systems facilitate the generation of digital elevation maps, enabling effective curb detection [40,41]. Balali and Golparvar-Fard [42] proposed an algorithm for the recognition of highway assets using video frames gathered from car-mounted cameras. In this approach, the performance is further enhanced through the utilization of motion cues and the preservation of temporal consistency.

Given the paper’s focus on point clouds, the literature review is primarily concerned with image-based point cloud data extraction. In the past two decades, point clouds have been widely used and by now considered as fundamental data entity as image data. In parallel, many methods applicable to imagery (2D) data have been extended as well as new methods have been created to process point cloud (3D) data.

Uslu et al. [43] proposed a 3D image-based object space reconstruction approach, using color, shape, and texture to perform automatic condition assessment for different highway assets. Although the initial findings are encouraging, clearly, more research and testing are needed. Balali et al. [44] used image-based 3D point cloud data to classify, detect, and localize traffic signs. While their method performs well, it is not applicable for road surface extraction. Gonçalves and Pinhal [45] presented the implementation of a mobile mapping system that uses action cameras mounted on the car. Despite the system being in development, a positional accuracy of approximately 30 cm has been achieved.

Frentzos et al. [13] developed a low-cost mobile mapping system using two machine vision cameras, dual frequency GNSS receivers, and low-cost IMU, demonstrating the efficient extraction of different road objects, such as road boundaries, road signs, and power poles. Their system is versatile and not focused on accurate road boundary extraction. Hasler et al. [46] provided a performance assessment of a smartphone-based mobile mapping for outdoors, achieving sub-decimeter relative 3D accuracy with a standard smartphone. Farhadmanesh et al. [10] investigated the feasibility of using photogrammetry for highway condition assessment. They performed a highway bridge, traffic sign, and pavement distress assessment, and the results were compared to mobile LiDAR technology for highway asset inventory, demonstrating the superiority of the MLS, yet photogrammetry could be an important and affordable alternative in certain circumstances.

Despite the promising results reported in earlier studies, there are still insurmountable and problematic situations in road extraction, necessitating further research and developments. Specifically, the identification of various road types, particularly roundabouts, remains a difficult issue, see [47]. The general trend is to utilize supervised and/or unsupervised methods to address the complexities and limitations encountered in accurately identifying and extracting diverse road structures [48].

This study aims to develop an approach that addresses existing gaps and overarching trends published in the literature. The proposed methodology is founded on unsupervised machine learning techniques, effectively mitigating the limitations associated with labeled data inherent in supervised methods. A crucial objective of this research involves the development of a versatile approach for identifying and extracting various types of roads surfaces and curbs. Another key goal is to establish a methodology that is universally applicable to data acquired from different mobile mapping systems (MLS and MMS). To the best of our knowledge, this study represents one of the initial endeavors in extracting road information from image-based point cloud data.

## 3. Methodology

The workflow diagram of the proposed methodology shown in Figure 1 has three major processing steps. In the initial stage, raw point cloud data obtained from MLS and MMS are filtered and then divided into ground and non-ground points. Next, the developed algorithms extract road boundaries and surfaces, based on the evaluation of elevation and slope. Finally, machine learning is utilized to eliminate incorrectly classified curb points and determine road boundaries.

### 3.1. Pre-Processing Steps

#### 3.1.1. Ground Filtering

3D point cloud data collected with MLS and/or MMS contain on-road points that reflected back from the road surface, curbs, road markings, sidewalks, manholes, and other road design elements, and then off-road points from objects, such as vehicles, traffic signs, light poles, trees, buildings, power lines, people, etc. Notably, the off-road points can be further divided into two groups of objects that are a part of the road infrastructure, such as traffic signs and light, light poles, and the rest of objects along the road corridor. The ground filtering process, also referred to as bare earth extraction, is a crucial step in the LiDAR data processing pipeline to separate the dataset into the ground and non-ground components.

In our approach, the filtered ground surface forms the continuous region that can support the movement of vehicles and pedestrians [49]. Following general practice, off-road points were identified and extracted from the ground point cloud data, and the remaining points were subsequently categorized into two groups: road and curb. Notably, curb points play an important role in defining the safe driving area of vehicles and delineating the boundary between road and off-road points [50].

We present a plane-based filtering approach to classify point clouds into ground and non-ground. The ground surfaces are filtered in an iterated process by plane fitting using first-order polynomials estimated in least squares regression. The initial step starts by dividing the 3D point cloud data into regular square grid where points with the minimum height in each block are retained [51]. Then, digital elevation models are generated by employing least squares fitting using the points in each block. Next, the height differences between all points in the point cloud with respect to the modeled surfaces, referred to as the height above ground (HAG), are computed. Non-ground points are expected to exhibit high HAG values; hence, setting a specific threshold value enables the identification/filtering of non-ground points. Then, the process was then continued to identify undetected non-ground points.

The initial weight values assigned based on the HAG values of the remaining points are adjusted based on the iterative weighting principle; points with a weight approaching 1 are classified as ground, whereas those with a weight approaching 0 are classified as non-ground. For this purpose, the Beaton–Tukey robust weighting function is employed [52]. In each iteration, a surface is fitted, and the deviations of each point from the surface are computed. Then, each point is assigned weights based on these calculated deviations using weight functions. The iteration continues until all classes are stabilized.

The proposed filtering approach efficiently filters ground points from the dataset; moreover, because ground points include potential curb points, they are utilized in the subsequent stages for curb point detection. Figure 2 shows a sample of the results of our proposed method, where the identified ground points are in red, other detected non-ground points are in black, and the surfaces created using ground points are depicted in blue.

#### 3.1.2. Extraction the Vehicle Trajectory from RAW LAS Data

Trajectory data, i.e., knowing the sensor location and/or pose, can support several downstream processing tasks and is frequently delivered as a supplementary dataset, providing valuable information, for example, for processing large amounts of point cloud data [53,54]. Furthermore, trajectory data are commonly utilized in various preprocessing steps, such as partitioning roads [28] or extracting tunnels or sectioning large datasets [55,56]. Furthermore, trajectory data can be used as vital auxiliary data in detecting off-road outlier/noise points [57] with road points serving as starting points [58]. Trajectory data are also employed in tracking targets [59], performing pavement surface distress analysis [60], or carrying out local coordinate transformations [61]. Typically, vehicle/platform trajectory information is not included in the product files and instead provided as a separate file. Unfortunately, such files may not be always available, due to a variety of reasons [62]. Therefore, if needed, a more general and inclusive approach to estimate trajectory from point cloud data are proposed here. The idea is to exploit the LiDAR point cloud data structure of the LAS format, which includes information about GPS time, optional RGB values, LiDAR intensity values, number of returns, scan angle, and coordinate information for each point.

Multibeam rotating laser scanners are predominantly used on mobile platforms and can be installed in any direction. Following the practice of airborne LiDAR, where the zero scan angle is defined by the vertical/nadir direction, the scan angle can be similarly defined in the mobile platform case. Notably, there is hardware-defined zero scan angle for every sensor, but it can be simply offset by any value as needed. Once the nadir looking scan angle is obtained, points with zero scan angle, *β,* are extracted and retained as points belonging to the sensor trajectory. There may be multiple points on each scan line with close to zero scan angle, and the GPS time in each scan line should increase near linearly in the direction of vehicle travel as seen in Figure 3.

In the next step, extracted points with close to zero scan angles are grouped according to GPS time. The point with the minimum GPS time is the first point, and then, all points are arranged in ascending order according to GPS time. Then, using an arbitrarily fixed time value, Δ*τ*, based on platform speed and scan frequency, the points are clustered into multiple segments between the min and max GPS times along the vehicle trajectory. The procedure is described by Equation (1).
(1)$$\left.\begin{array}{c} P_{\left(\beta \right)}{(t}_{2\_\mathrm{gps}})\;>{\;P}_{i\left(\beta \right)}(t{\_}_{\mathrm{gps}})\;>\;P_{\left(\beta \right)}{(t}_{1\_\mathrm{gps}})\;\\ P_{\left(\beta \right)}{(t}_{2\_\mathrm{gps}})=P_{\left(\beta \right)}{(t}_{1\_\mathrm{gps}})+\Delta \tau \end{array}\right\}T_{i}$$
where *Ti* is approximated trajectory data, $P_{i\left(\beta \right)}$ is possible trajectory points with zero scan angle, *β* is zero scan angle, Δ*τ* is time threshold value, $P_{\left(\beta \right)}{(t}_{1\_\mathrm{gps}})$ is first GPS time threshold, and $P_{\left(\beta \right)}(t_{2\_\mathrm{gps}})$ is second GPS time threshold.

The centroid of each cluster using Equation (2) is computed to estimate trajectory points. Figure 4 shows the process of defining the trajectory polyline based on the extracted vertices. Notably, the trajectory line can be further smoothed if needed.
(2)$$\bar{x}=\frac{\sum_{i=1}^{n}x_{i}}{N}\bar{y}=\frac{\sum_{i=1}^{n}y_{i}}{N}$$
(3)$$V_{i}=({\bar{x}}_{i},{\bar{y}}_{i})$$
where $\bar{x}$ and $\bar{y}$ are centroid coordinates of each cluster, and *x_i_* and *y_i_* are the coordinates of each *N* points in the cluster. The trajectory information obtained from the MLS data were also used for the road extraction process based on the MMS data, as both datasets were collected along the same test route. Naturally, the camera pose data can be used in the creation of trajectory information in cases where only MMS is used.

### 3.2. Road Boundary Detection and Road Surface Extraction

#### 3.2.1. Partitioning Point Clouds into Sections

The detection of road boundary points is carried out on smaller segments (B_w_) using the estimated trajectory; this is mainly performed for efficient implementation purposes, such as reducing CPU time. Since the orthogonality of the scanlines with respect to the trajectory cannot be guaranteed, the clusters are refined into segments that represent points in a rectangle that is generally perpendicular to the trajectory line. During this process, absolute coordinates are used, and the process delivers the results in the local mapping frame.

Figure 5 shows the process of calculating vertices for each segment using trajectory points *V*_1_ and *V*_2_. These points could be the estimated vertices or can be interpolated based on the trajectory. The heading angle *θ* computed from *V*_1_ and *V*_2_ defines the road direction, which is used to estimate the corner points of each segment, i.e., *V*_11_ and *V*_12_ and then *V*_21_ and *V*_22_, based on Equations (4) and (5). The width, *w*, of each block is user-defined, while the length, *l*, corresponds to the distance between previously identified vertices.
(4)$$V_{i}X=P_{i}X+w\times \cos(\alpha )$$
(5)$$V_{i}Y=P_{i}Y+w\times \sin(\alpha )$$

Figure 6 shows the obtained block formed by the segments and a sample cross-section (CS_w_) recovered from one of them. Cross-sections were extracted from each segment using this processing step, and curb points were calculated inside these cross-sections using the approach explained in the subsequent section.

#### 3.2.2. Extracting Road Boundaries Using Slope and Elevation Threshold

The road boundary detection from point cloud starts with searching for possible curb points. In general, curbs are boundaries separating roads and other surfaces. Hence, the difference in spatial characteristics on the two sides of the boundary can be exploited to determine the initial curb line. As indicated by Yang et al. [28], there are three major types of curbs, and based on their characteristics, we used two indicators, elevation and slope change, to detect curb points. Normally, curbs are parallel to the road direction, while sudden slope and elevation changes are almost always perpendicular to the road surface, see the sample road section in Figure 7.

The detection of road boundaries or coarse curb points is performed on the cross-sections extracted from each segment, involving three distinct stages: the down-sampling of the dataset, the detection of curb points, and the detection of non-curb points. In the initial stage, datasets within each cross-section are down-sampled to improve the processing speed of the curb detection process. Thus, the dataset is divided into a voxel grid determined by the user, and the height value assigned to a voxel is the average of the heights of the points within that voxel. With the selection of a suitable voxel size, the structure of the point cloud is well preserved.

In the second and third stages of the process, the curb and non-curb points are detected. The points along road edges typically exhibit height differences of 5–30 cm depending on the local regulations in the region when compared to points on pavement surfaces. Additionally, while there are typically fewer points on the near-vertical road edge surfaces, the sudden change in slope between adjacent points around the road boundary is prevalent. Therefore, by leveraging these two criteria, height differences and slope change, it is possible to distinguish between a curb and a non-curb point. To account for this variation, we employ the following formula:H_min_ ≤ ΔH_i_ ≤ H_max_(6)
where Δ*H_i_* is the maximum height difference between candidate curb points and neighborhood points, and *H_min_* and *H_max_* are the minimum and maximum curb height difference, respectively.

If points satisfy the conditions given in Equation (6), they are labeled as candidate curb points, otherwise they are classified as non-curb points. Next, the slope-based constraint is applied to detect curb points. In each generated cross-section, the search algorithm runs from the center to both outward directions, and the slope and height difference between adjacent points is evaluated, see Figure 8.
(7)$$S_{\mathrm{pi}}=\tan\theta =\frac{z_{2}-z_{1}}{\sqrt{(x_{2}-x_{1}{)}^{2}+(y_{2}-y_{1}{)}^{2}}}$$
where $\left(x_{1},y_{1},z_{1}\right)$ and $\left(x_{2},y_{2},z_{2}\right)$ are the coordinates of adjacent points in each cross-section, and $S_{\mathrm{pi}}$ refers to the calculated slope value between consecutive points. If the slope value $S_{\mathrm{pi}}$ of a point is greater than the slope threshold $S_{t}$, and at the same time, height difference of the same point is within the specific range of curb height $\left[H_{\min},\;H_{\max}\right]$, the point is classified as a curb point.

#### 3.2.3. Road Boundary Tracking and Refinement

Data gaps in road surfaces are frequent due to occlusion by a range of obstacles, including vehicles, street furniture, pedestrians, etc., in normal traffic conditions, resulting in undetected curb points [63]. In addition, certain points with geometric features similar to curb points could be mislabeled as curb points. Therefore, filling the gaps and removing erroneous non-road surface points are important to obtain an optimal road surface representation, which can support efficient extraction, tracking and refining road boundaries and removing faulty curb points [28].

Here, a machine learning-based methodology for tracking and refining curb points is proposed. The method is based on DBSCAN, which is a machine learning method suitable for utilizing the curb points detected in the preceding step to determine and eliminate incorrectly identified curb points. Subsequently, the proposed approach classifies road edges as straight and curved lines, and then, RANSAC is utilized to eliminate outliers in the classified curb points. Finally, gaps among the curb points are filled based on the collinearity condition, and then, the B-spline algorithm is employed to generate smooth boundaries.

Applying DBSCAN for incorrectly detected curb points: The DBSCAN algorithm is an unsupervised clustering technique intended to cluster spatial datasets based on density information, which is used to identify false curb points in our workflow [64]. DBSCAN does not require a pre-specified number of clusters, treats outliers as noise instead of assigning them to a separate cluster, and overcomes the limitations of conventional clustering algorithms like k-means by effectively clustering data groups with varying sizes and shapes [65].

The DBSCAN method requires two control parameters, i.e., the minimum number of points, minPts, and the radius value, ε, to control the clustering process. The basic steps of the algorithm are as follows:A random point, X, is chosen from the LiDAR point cloud, and then, points are examined based on the parameters minPts and ε. The distances from the selected point X to all other points are calculated. Within this framework, all neighbors of point X, whose distance from X is either less than or equal to ε or falls within the specified radius (eps − ε) of the circle, are retained.The point X is considered a core point if there are more points than the specified minPts within the space around it defined by the given ε threshold. If it has less than the minPts in the ε-neighborhood, then it is identified as a border point. Finally, if the point is not classified as a boundary or core point, it is considered a noise point. In Figure 9a, points X, Y, and Z are selected as core points within the ε-neighborhood, satisfying the condition minPts > 8.The DBSCAN method creates clusters based on the density-reachable and density-connected concepts. Points that are within a chain of distance ε are considered density-reachable and density-connected; notably, point Z can be density-reachable from point X. Thus, this method creates clusters by connecting core points and their neighbors in dense regions within a distance of ε [66].

This iterative process continues until the point groups are stabilized [65,67]. In general, the minPts value is determined through trial and error by utilizing familiarity with the dataset. Following the determination of the minPts value, the ε value is automatically ascertained using the approach recommended by [68]. This technique computes the average distance between each point and its k nearest neighbors, where the parameter k is defined as the selected minPts value. Afterward, the average k distances are then plotted in ascending order on a k-distance graph. The inflection point of the curve, commonly referred to as the “knee”, is identified where the graph exhibits the greatest slope. The optimal value for ε can be determined at this point of maximum curvature. The implementation is available in Matlab 2022.

The candidate curb points, identified through the process described in Section 3.2.2, are clustered using the DBSCAN approach, and the detected false curb points are classified as outliers and removed as seen in Figure 9b,c.

Detection of curb boundaries: A contour-based approach developed by Awrangjeb [69] is employed for the identification and delineation of curb point boundaries. Originally developed for the extraction of outer building boundaries, the method employs the Delaunay triangulation approach. However, due to its success in extracting the boundaries of point cloud data in different shapes and structures, it was adopted for the extraction of boundaries for the candidate curb points. The obtained boundary points are used to partition the data into sections in the next stage, see Figure 10.

Identification of curve and straight curb sections: An initial step of the refining process is the segmentation of the curb line into curved and straight sections. First, the curb points, identified in the previous stages, are partitioned into sections with the aid of the boundary points, as shown in Figure 10a. Next, lines are fitted to the points in each section using Equation (8), and then, the direction of each line is computed using Equation (9). By inspecting the changes in the direction of the line across consecutive segments using Equation (10), the pavement sections are categorized as either straight or curved. To achieve this, the direction difference values are compared to pre-defined threshold values. The threshold value was experimentally determined as 15°, and segments are classified as straight if the direction difference is below the threshold or as curved if it exceeded the threshold.
y = m × x + c (8)
(9)$$m=\frac{n\sum \mathrm{xy}-(\sum x)(\sum y)}{n\sum x^{2}-(\sum x{)}^{2}}$$
Δm = m_i+1_ − m_i_(10)
where *x* and *y* correspond to the coordinates of the points within each section, *m* represents the direction, *n* denotes the number of points, *c* signifies the intercept, and Δ*m* stands for the slope difference values. In the following section, comprehensive explanations pertaining to the refinement of curb points within the identified straight road segments are provided.

Here, a novel approach is introduced for extracting curb points within the determined curved sections of the road. In this approach, circles (*ki*, *ki* + 1, …) with a predefined radius are generated in curved sections, starting from the beginning of the detected curve. In sequential processing, least squares are used to fit a line to the points inside the circle placed at the start of the curve. The closest point to the intersection of the fitted line and circle defines the center of the next circle. This process is repeated, including seed points, circles, and line fitting, until the curved part is finished as illustrated in Figure 10b. The center points of the circles represent the final results, i.e., the best reconstruction of the curved curb line. Figure 10d shows the refined curb point representation, superimposed on the original point cloud. Using B-spline fitting, the smooth road boundary is created, as depicted in Figure 10e.

Road edge refinement: This process constitutes the last phase of the tracking and refinement process. After the above-discussed processing steps, slight deviations may be observed in some parts of the extracted curb. While offsets from the actual boundary are generally small, improvements to these outlier points are necessary to increase the accuracy and robustness of road boundary determination. Consequently, in a critical final step, the RANSAC algorithm is applied to eliminate outliers and further optimize the precision of the boundary detection process. RANSAC is a widely used algorithm introduced by Fischler and Bolles [70] that offers a simple, efficient, and robust method for estimating parameters in the presence of noise and outliers. The algorithm operates by randomly selecting sets of data points and iteratively constructing models based on these sets. In each iteration, the algorithm identifies inlier points that conform well to the candidate model by rigorously testing against the remaining data. A score value is then calculated for the inlier points, and the process is repeated for a specified number of trials. Ultimately, the model with the highest score and the most inliers is selected.

## 4. Experiments

### Study Areas, Data Acquisition Systems, and Datasets

Two datasets acquired with MLS and MMS were used for assessing the performance of the proposed method. The test site is located at Yildiz Technical University Campus, Istanbul (410 01′ 47.7851″ N, 28 53′ 24.7852″ E). Figure 11 shows the test area outlined in red. This region extends from the north to the northeast of the campus and has a total road length of 2.2 km, and a divided two-lane road of 1.1 km with one-way traffic in both directions. The MLS system generated about 62 million points with an average density of 800 pt/m^2^. In comparison, the MMS system created a total of around 83 million points with 1000 pt/m^2^ density; notably, the modest point density option was used in the photogrammetric point cloud generation.

The overall road boundary comprises approximately 4.4 km when accounting for the right, middle, and left road boundaries. It is a wide street, a typical urban road environment, which includes road surface, traffic signs, light poles, power lines, trees, low vegetation, buildings, moving and parked vehicles, and pedestrians. Furthermore, the road has straight sections and a mixture of regular and irregular curbs alongside multiple lanes that add to the complexity of the road environment, and thus, it is very useful for the comprehensive testing of the proposed method. All the algorithms are implemented in the Matlab environment, and the processing was performed on a personal computer equipped with a 2.50 GHz Intel(R) Core (TM) i7-4710 processor and 16 GB RAM.

The MLS dataset was acquired by the Phoenix Alpha AL3-32 LiDAR system (https://www.phoenixlidar.com/scout-32/ (accessed on 4 October 2023), mounted on the rear of a pickup truck, as shown in Figure 12. It is a lightweight and long-range LiDAR system that is equipped with a Velodyne HDL-32e LiDAR sensor that is oriented at a pitch angle of 45° with respect to the driving direction. The laser scanner generates 700,000 laser points per second and has a field of view scan range of about 360° and 40° horizontal and vertical directions, respectively. The navigation system consists of an OEM high-precision IMU, NovAtel OEM6 dual-frequency GNSS receiver, and a microcomputer. To allow for PPK solution, a GNSS base station was installed in the test area. LiDAR data processing was conducted through the utilization of LiDARMill [71]. NavLab, a crucial feature of LiDARMill, was used to integrate the IMU and GNSS data, generating a precise and accurate trajectory that was used to create the LiDAR point cloud.

The MMS platform, shown in Figure 13, was equipped with two GoPro Hero 7 action cameras, Xsens MTi-G-700 IMU and two Topcon HyperPro dual-frequency GNSS receivers; all mounted on the roof of a van. The sampling rate of the cameras, GNSS receivers, and IMU measurements were 30 FPS, 10 Hz, and 400 Hz, respectively. The video resolution of MMS during data acquisition was 2704 × 1520. GNSS/IMU data were processed using Inertial Explorer by NovAtel and, subsequently, used for georeferencing images and point cloud generation. The system calibration, time synchronization, georeferencing of frames, and detailed system analysis can be found in [71,72].

## 5. Results

The various parameters used to control the entire processing workflow are listed in Table 1.

The road boundary extraction results, represented by red points, are overlaid on the MLS and MMS point clouds, respectively, as illustrated in Figure 14. For better visualization, multiple sections of the road segments are enlarged to show road boundary extraction performance, as depicted in Figure 15.

The results obtained using the proposed methodology on straight and curved roads are presented in Figure 15a,b,g,h, indicating the method’s effectiveness in general road boundary extraction. Moreover, the proposed method demonstrates its capability to accurately deduce road boundaries at intersections and curves, as illustrated in Figure 15c,d. Additionally, our proposed methodology proficiently captures and defines road boundaries even in complex intersections. Notably, the road boundaries of roundabouts and traffic islands of various shapes are well defined, as observed in the results Figure 15d.

However, there have been instances where the determination and extraction of road boundaries were found to be inadequate and inconsistent. As illustrated in Figure 15b,g,h,j,k, the lack of a discernible change in curb height from the road surface resulted in a failure to identify curb points, consequently leading to incomplete road boundaries in these particular sections.

Additionally, as depicted in Figure 15e,f, the other side of some road medians pose challenges in determining the road boundaries. The inability to acquire data from these curb surfaces, which are on the opposite side of the vehicle’s direction of travel, makes it even more difficult to determine the road boundaries in these regions. Additionally, as seen in Figure 15c, the detection of curb points in curved areas has also proven to be challenging. It was observed that road boundaries could be extracted with greater precision when the trajectory of vehicles aligned more closely with the road edge lines. To mitigate both situations, multiple scans from various perspectives must be acquired to obtain comprehensive point clouds of the area.

In Figure 16, images show the areas with missing road boundaries (regions represented by Figure 16h,j,k). In Figure 16a, the curb points have low heights, and the road surface is uneven; thus, the curb surface is almost at the road level. Figure 16b shows a case when the pavement heights are very low. Figure 16c shows a median head in the study area that has undergone deformation and is almost leveled to the surrounding road. Due to all these factors, road boundaries could not be extracted in these regions from none of the datasets.

To quantitatively assess the results, about 2600 reference points along the road were surveyed using geodetic GPS receivers in the RTK mode. In order to obtain a reliable representation of the road boundary, the interval between the reference points was set at 0.5–1 m for curved sections and 2–2.5 m for straight sections. The accuracy of 95% of the reference points fell within the range of 1–4 cm.

The numerical evaluation of road boundaries extracted from MLS and MMS data were based on comparing the extracted curb edges with reference data. Road boundary points were classified as True Positives (TP) if the point lies within a 5 cm buffer zone around the corresponding reference points. Otherwise, these points were evaluated as False Positive (FP). This statistical evaluation provided a representative analysis of the road boundary determination capabilities of the proposed methodology [15].

The three commonly employed metrics, i.e., completeness, correctness, and quality were computed, allowing the assessment of the effectiveness of the proposed road boundary extraction method. These metrics are defined as follows:(11)$$\mathrm{Completeness}=\frac{\mathrm{TP}}{L_{r}}$$
(12)$$\mathrm{Correctness}=\frac{\mathrm{TP}}{L_{e}}$$
(13)$$\mathrm{Quality}=\frac{\mathrm{TP}}{L_{e}+\mathrm{FN}}=\frac{\mathrm{TP}}{\mathrm{TP}+\mathrm{FP}+\mathrm{FN}}$$
where L_r_ and L_e_ are the total lengths of ground-truth road boundaries and extracted road boundaries, respectively, and TP, FP, and FN are the lengths of correctly or falsely detected or undetected road boundaries, respectively.

Table 2 and Table 3 present the final results for both datasets. These findings indicate that our proposed method exhibits a high accuracy in extracting road boundaries from MLS data but yields a somewhat lower performance with MMS data. The reason for this disparity lies in the fact that point cloud data obtained from mobile images can be highly impacted by various factors, including sensor limitations, such as blurring and big scale changes, environmental conditions, such as shadows created by objects like buildings, trees, and vehicles, and processing errors. Additionally, adequate illumination is necessary, as excessive sunlight may cause reflections that introduce further distortions in the images. Notably, the professional grade Velodyne scanner costs an order more than the consumer grade GroPo camera, so in that sense the MLS performance is quite reasonable.

In summary, the point cloud data derived from mobile sensors may suffer from reconstruction errors and contain gaps, noise, and clutter. In addition, we observed that photogrammetric point cloud data contained more errors than MLS data. It is essential to acknowledge that there are various types of cameras available in the market that differ in terms of image quality and performance with proportional price.

## 6. Conclusions

This study introduces a novel methodology, integrated with machine learning for the rapid, highly automated, and precise extraction of road boundaries from MLS and MMS point cloud data. Extensive testing has demonstrated that the proposed method exhibits high performance in extracting and modeling 3D road boundaries with accuracy levels exceeding 98% (MLS) and 93% (MMS), respectively.

The MLS and MMS systems exhibit distinct benefits and drawbacks when compared to each other. The primary advantage of the MLS system over MMS lies in its better immunity to environmental factors, including lighting and weather conditions. Moreover, MLS systems directly generate 3D coordinates at a typically faster rate than MMS. Despite yielding a lower point density than MMS, MLS comes with higher software and hardware costs. Nonetheless, both systems can be deployed on vehicles and can be utilized to cover long distances quickly. Notably, MMS requires a lower investment cost in comparison to MLS. In general, selecting between image-based and LiDAR-based mobile mapping systems depends on the particular requirements of the application, including target accuracy and performance as well as available budget.

The investigation indicated that MMS has the potential to emerge as a viable alternative to MLS systems as well as to traditional surveying techniques. Furthermore, given the potential of image-based systems to extract road information, future case studies may incorporate experiments with various imaging sensors and multiple sensor configurations.

## Figures and Tables

**Figure 1 sensors-24-00503-f001:**
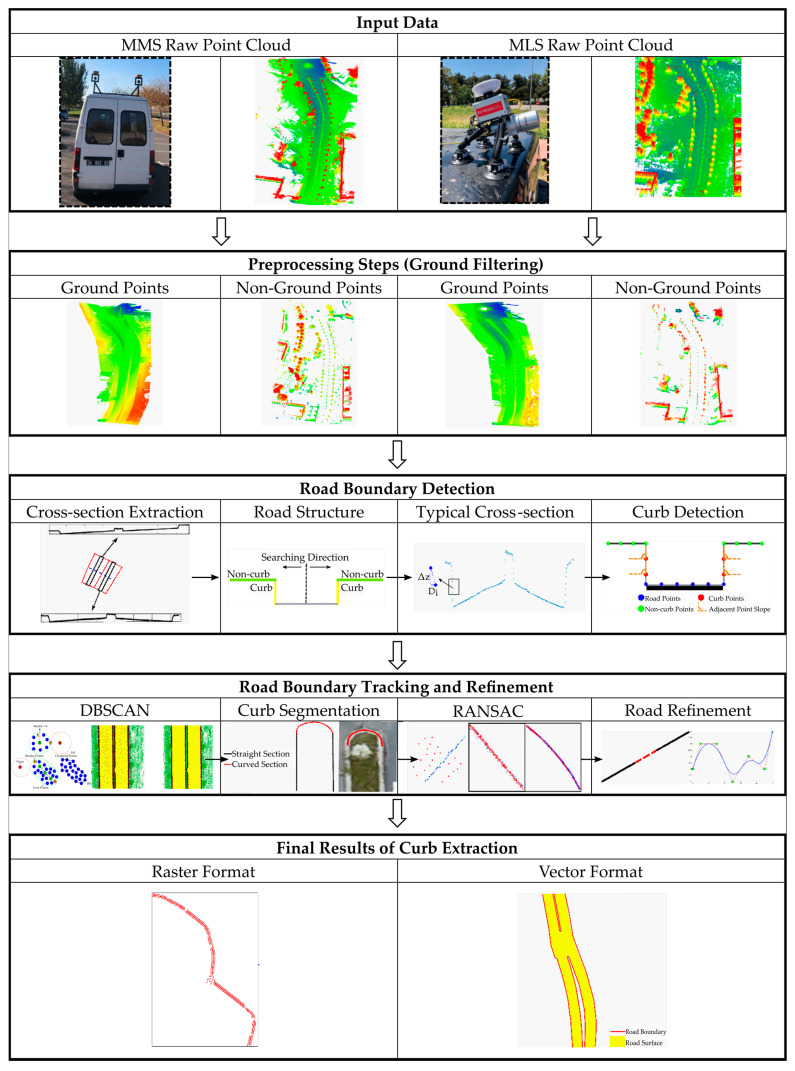
Workflow diagram for the extraction of road parameters.

**Figure 2 sensors-24-00503-f002:**
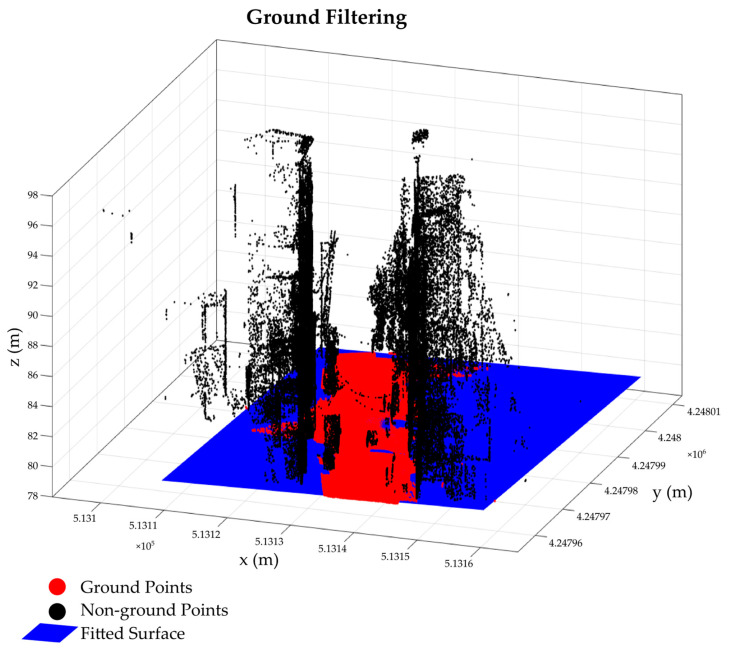
An example of filtering results.

**Figure 3 sensors-24-00503-f003:**
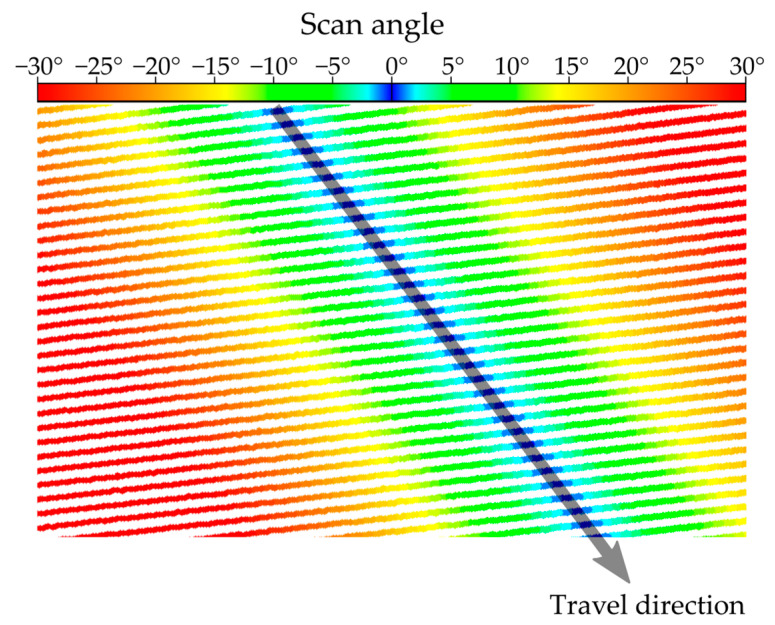
Scan lines with scanning angle range of ±30°; the points with close to zero scan angle are highlighted in blue, and the arrow shows increasing GPS time and vehicle travel direction.

**Figure 4 sensors-24-00503-f004:**
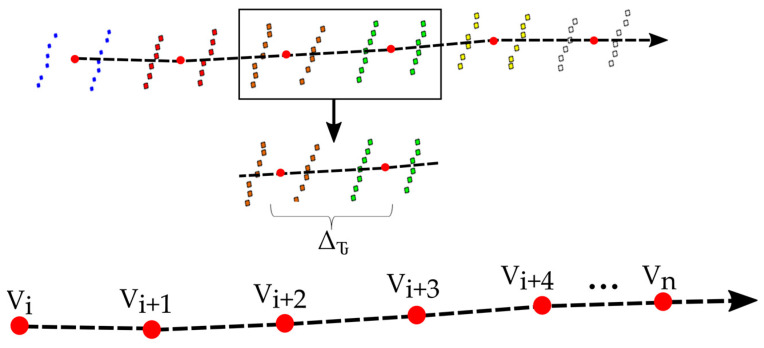
The scan lines shown in different colors were grouped based on a time threshold, and the vertex coordinates ($V_{i})$ of the trajectory polyline are calculated.

**Figure 5 sensors-24-00503-f005:**
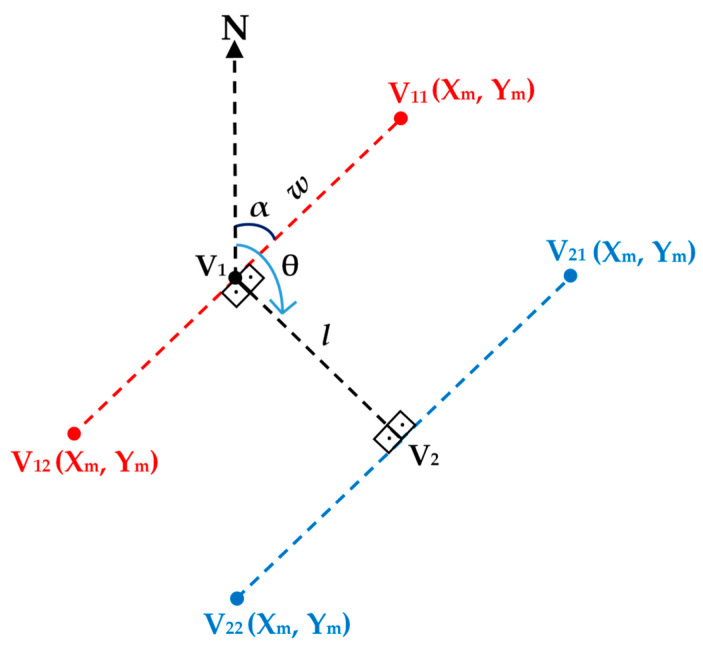
The process of calculating vertices of each segment.

**Figure 6 sensors-24-00503-f006:**
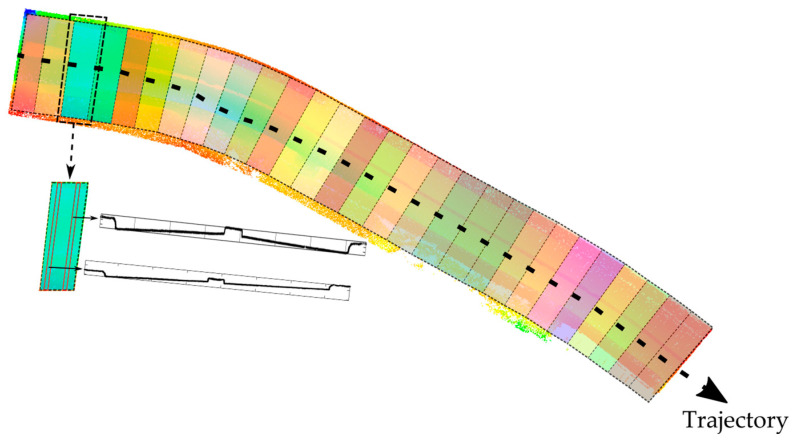
Segmentation and cross-sectioning of the test area based on trajectory. Each segment is shown in a different color.

**Figure 7 sensors-24-00503-f007:**
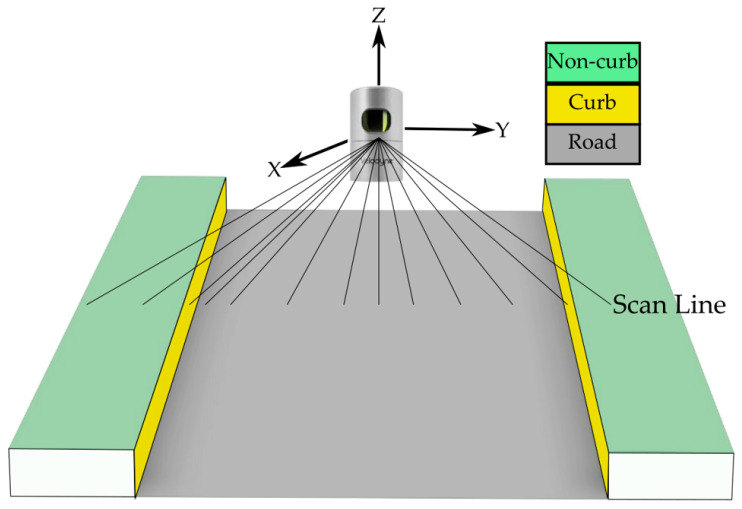
Typical road structure.

**Figure 8 sensors-24-00503-f008:**
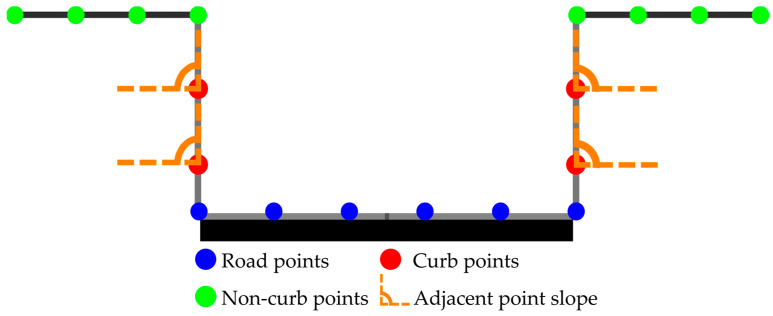
Curb detection procedure: blue points represent areas where height and slope thresholds are not met, green points signify locations meeting only the height threshold, and orange points meet both height and slope criteria, classifying them as curbs.

**Figure 9 sensors-24-00503-f009:**
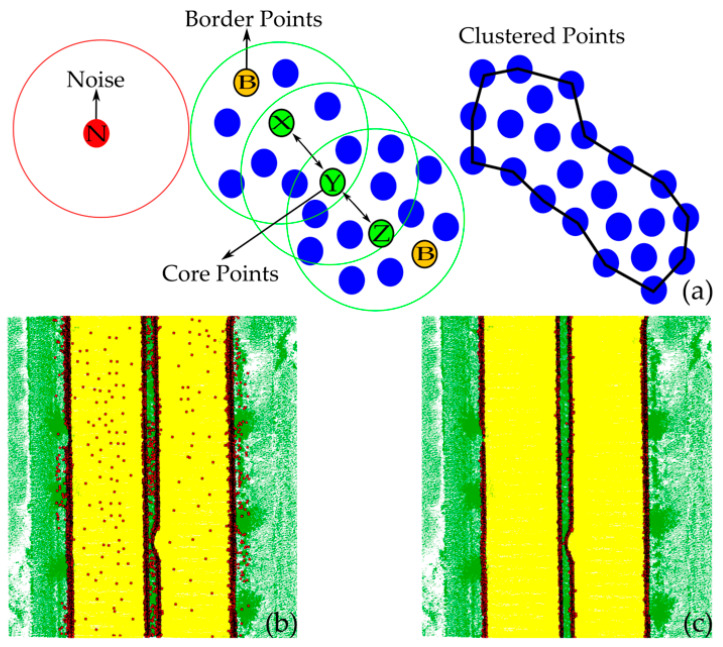
(**a**) The basic steps of the DBSCAN algorithm, (**b**) extracted candidate curb points, and (**c**) identification and removal of incorrectly detected curb points from the dataset.

**Figure 10 sensors-24-00503-f010:**
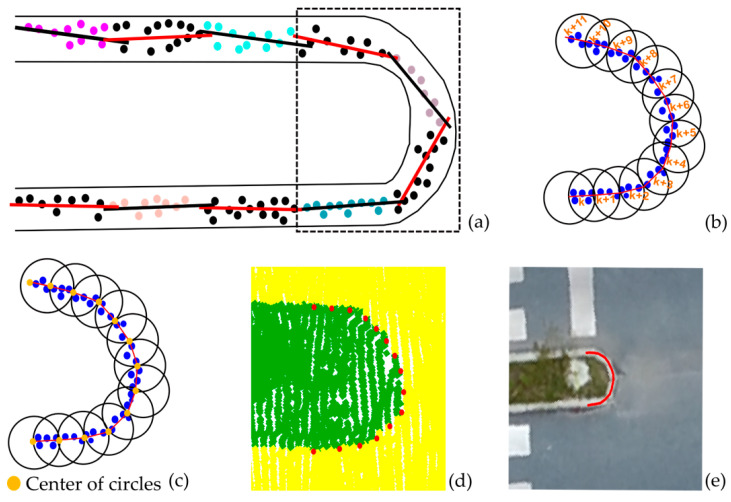
(**a**) 2D view of the classified road curb points with color-coded partitioning and lines fitted in each segment, (**b**) detected curved sections with generated circles, (**c**) circle centers marked in orange, (**d**) circle center point-defined curb line superimposed on point cloud, and (**e**) smooth boundary line.

**Figure 11 sensors-24-00503-f011:**
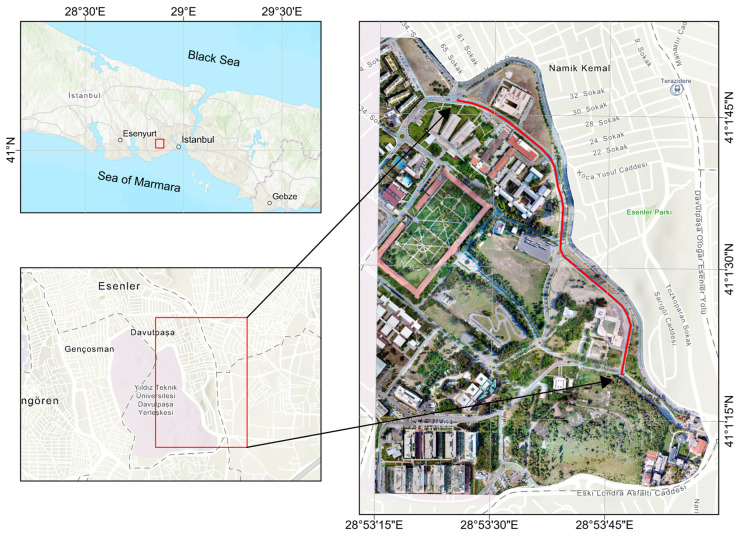
The left side displays the region containing the university campus, while the right side shows an orthophoto of the campus with the test route marked by a red line.

**Figure 12 sensors-24-00503-f012:**
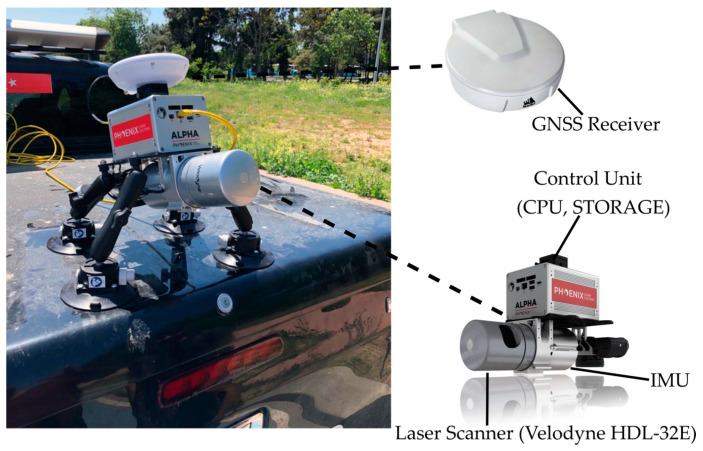
The Phoenix Alpha AL3-32 LiDAR system mounted on a pickup truck.

**Figure 13 sensors-24-00503-f013:**
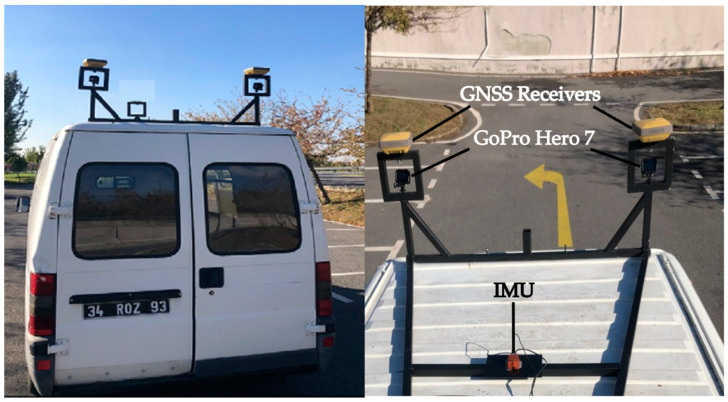
MMS data acquisition system and sensor configuration.

**Figure 14 sensors-24-00503-f014:**
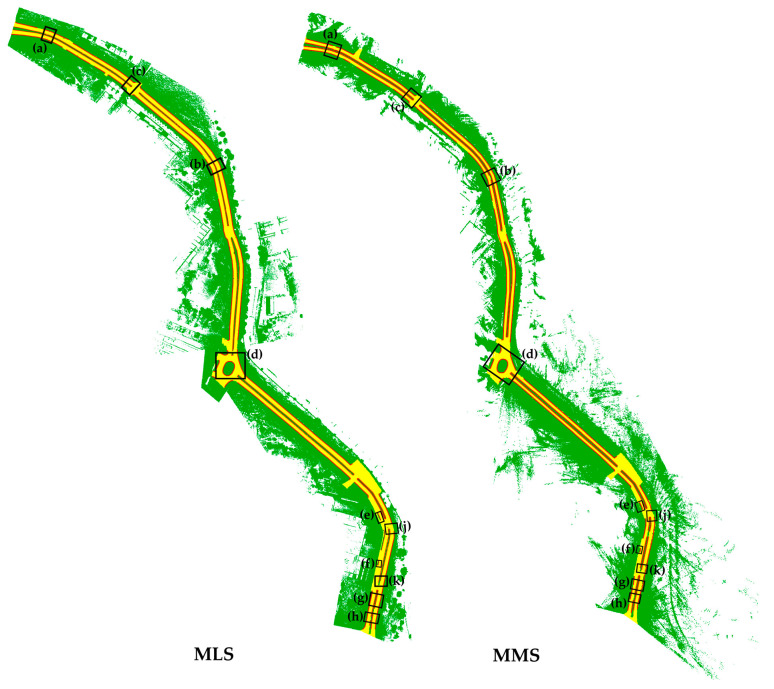
The detected road boundaries of the entire area: MLS and MMS.

**Figure 15 sensors-24-00503-f015:**
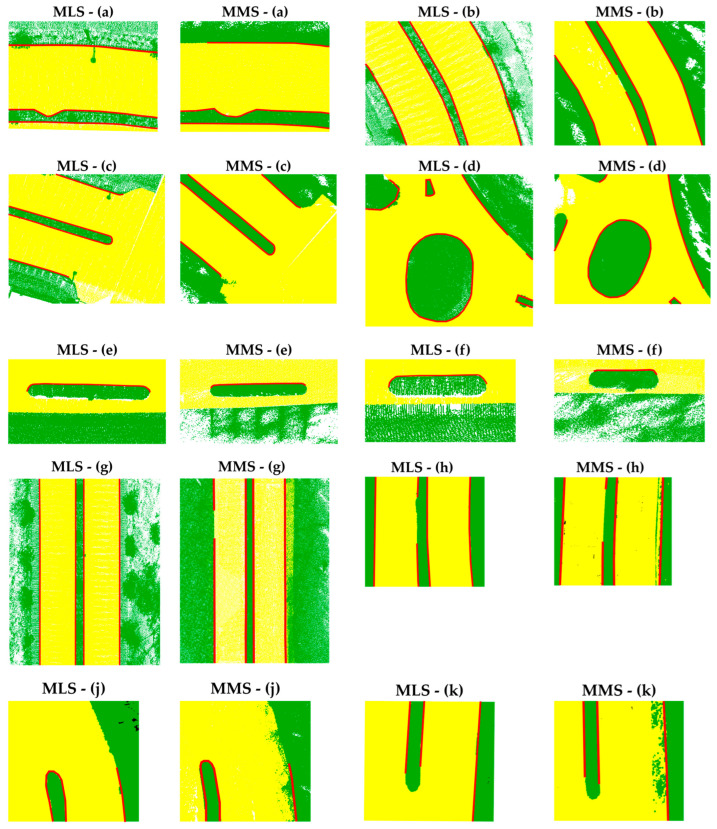
Side-by-side comparison of various road sections for MLS and MMS: (**a**) complex curb line, (**b**) curved road, (**c**) road intersection and turning, (**d**) road roundabout, (**e**,**f**) two different road medians, (**g**,**h**) straight road, (**j**) road median and curved road, and (**k**) road median and straight road.

**Figure 16 sensors-24-00503-f016:**
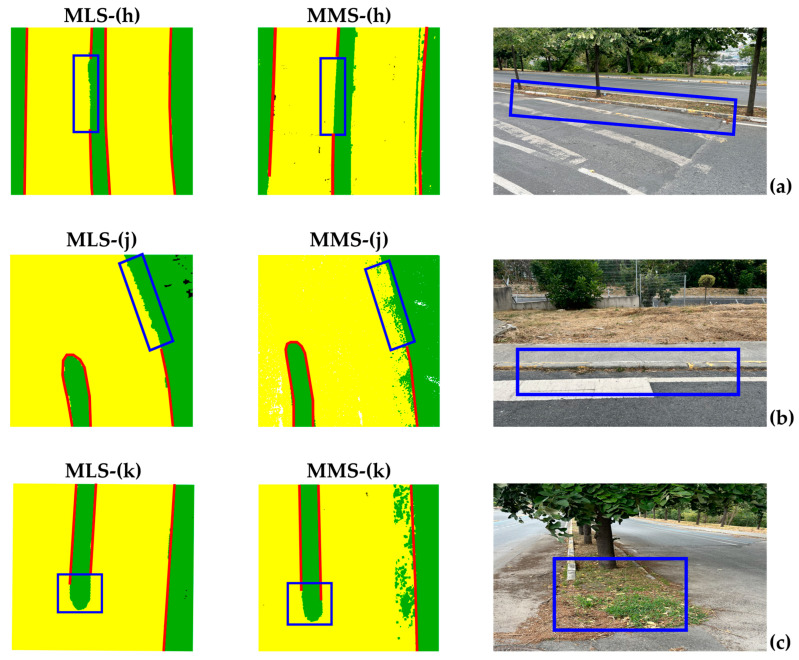
Example of missing road boundaries extracted from MLS and MMS data, (**a**) low curb height and bump overlap, (**b**) low curb height, and (**c**) deformation of the median head.

**Table 1 sensors-24-00503-t001:** The parameter settings of the proposed algorithm as applied to the two datasets.

Parameters	Description	Values	
		MLS Dataset	MMS Dataset
Δh	Height difference threshold for ground filtering	0.2 m	0.2 m
B_w_	The segment width	5 m	5 m
CS_w_	The cross-section width	0.02 m	0.02 m
S_t_	Slope threshold	45°	45°
H	The elevation difference for detecting road curbs	H_min_ = 0.05 m, H_max_ = 0.30 m	
Epsilon (ε)	Radius value for searching minimum point	Automatic	Automatic
MinPts	Minimum number of points	7	6

**Table 2 sensors-24-00503-t002:** Road boundary extraction results.

	L_r_ (m)	L_e_ (m)	TP (m)	FP (m)	FN (m)
MLS	4052.2	3832.2	3777.5	54.6	187.7
MMS	4052.2	3658.1	3422.8	235.2	395.0

**Table 3 sensors-24-00503-t003:** Quantitative evaluation results.

	Completeness (%)	Correctness (%)	Quality (%)
MLS	93.2	98.6	93.9
MMS	84.5	93.6	84.5

## Data Availability

The data are not publicly available due to privacy restrictions.

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
