# Peer review of "3D Road Boundary Extraction Based on Machine Learning Strategy Using LiDAR and Image-Derived MMS Point Clouds"

_sensors, 2024, doi:10.3390/s24020503_

Round 1

Reviewer 1 Report

Comments and Suggestions for Authors

The paper presents a new method to automatically extract 3D road boundaries from point clouds derived from image and LiDAR sensors, deployed on mobile mapping platforms. Two experiments were carried out on two datasets acquired with MLS and MMS to assess the performance of the proposed method.

The paper is reasonably organized and well written with a robust terminology.

Some comments below:

The experiments nicely show the results obtained by the proposed method, but the evaluation is done only in terms of quality of output data. There is no comparison with other methods, so it is very hard to tell, whether the accuracy advantage of the proposed method.

Efficiency analysis on the proposed method was not conducted in the experiment. The proposed method has many steps, but no ablation experiment was conducted.

There are many parameters used in the proposed method. How the parameters are determined. The minimum number of points of used for MLS and MMS is different (other parameters for the two datasets are the same), why?

Author Response

The authors would like to thank the reviewers for their thorough work, which greatly helped us improve the manuscript.
Some comments below:
The experiments nicely show the results obtained by the proposed method, but the evaluation is done only in terms of quality of output data. There is no comparison with other methods, so it is very hard to tell, whether the accuracy advantage of the proposed method.
Answer: This is very valid point, and the fact is that the authors tried to find benchmarks suitable for testing the developed algorithm. We have looked at the ISPRS, especially Geo Bench listed benchmark dataset but found no suitable ones. Therefore, the proposed method in this study was only tested on our self-collected dataset. While some studies in the literature compare the results based on specific datasets with the results of different methods, it is important to note that, in almost all cases, the developers of the different methods were invited to test their approaches on the dataset specified in the paper. Unfortunately, we had no such an opportunity. Furthermore, we wanted to test the performance of proposed method on point clouds acquired at the same site by LiDAR (MLS) and photogrammetrically (MMS), which left no other option than acquiring our own data. Finally, while it is true that many studies in the literature rely on self-collected data, depending on the objectives, there is a limited number that utilizes public datasets, such as the Kitti dataset.
Efficiency analysis on the proposed method was not conducted in the experiment. The proposed method has many steps, but no ablation experiment was conducted.
Answer: Our primary goal was to develop a method that achieves the best possible performance for the given problem without a specific focus on creating a computationally efficient approach. While computational efficiency is undoubtedly important, it was not the primary consideration in this phase. We acknowledge its significance and intend to address this aspect in subsequent stages of our research, particularly as we extend our evaluations to include other datasets for comprehensive performance assessments.
There are many parameters used in the proposed method. How the parameters are determined. The minimum number of points of used for MLS and MMS is different (other parameters for the two datasets are the same), why?
Answer: As usual, various parameters were employed in the application of the proposed method, including segment width (Bw), cross-section width (CSw), slope threshold value (St), elevation difference for curbs (Hmin, Hmax), minimum number of points (MinPts), and Epsilon (e) value. In addition, to comply with road design regulations and existing information for road design, such as curb height, etc., three parameters—specifically Hmin, Hmax, and St—were determined. Since both datasets were obtained from the same test route, the minimum and maximum curb heights, as well as the slope change on the curb surface, these values were set to be identical. The values for parameters like Bw and CSw were determined through experimentation. Clearly, the characteristics of the study area was playing a role in determining these parameters. The MinPts parameter varies significantly based on the dataset and was manually determined through running numerous tests on the dataset to determine the most suitable parameter. Note that all parameter values are provided in the table for both data sets.

Reviewer 2 Report

Comments and Suggestions for Authors

This  study proposes  a  new  method  to  automatically  extract  3D  road boundaries from point clouds derived from image and LiDAR sensors, this method was tested and validated using two data sets, however this paper has the following issues that require further modification:
1) DBSCAN and RANSAC are relatively traditional and classic road boundary segmentation and identification methods, so the understanding and technical innovation of the paper need to be further clarified;
2) The paper claims that the completeness rates were 93.2% and 84.5%, while the correctness rates were 98.6 and 93.6%, respectively, but the effectiveness and accuracy of this method were not compared with other methods;
3) How time-consuming and practical the engineering calculation of the proposed 3D road boundaries acquisition method is, the paper lacks corresponding verification;
5) There are some unclear or wrong expressions in the paper, such as the meaning of HD in the abstract, etc. and grammatical errors that need to be checked;

Comments on the Quality of English Language

grammatical errors that need to be checked;

Reviewer 3 Report

Comments and Suggestions for Authors

Thank you to the authors for presenting such an interesting piece of work. The proposed approach demonstrates high performance in extracting road boundaries and surfaces from point clouds. The experimental results confirm the success of the proposal. However, before publishing this work in the Journal of Sensors, several areas need improvement. My detailed concerns are outlined below:

1.      Include the objective of the work and the research gap or scientific problems related to this topic in the abstract.

2.      The 'Related Studies' section classifies road boundary extraction methods into activity-driven, feature-driven, and model-driven methods. There appears to be little distinction between feature-driven and model-driven methods. Please reconsider the categorization rules and provide clearer descriptions for these categories at the beginning of the section.

3.      The initial sequencing in the 'Related Studies' section omits activity-driven methods and starts with model-driven methods. This lacks coherence and needs explanations.

4.      Placing the description of activity-driven methods at the end of the review of model-based methods seems illogical.

5.      The organization of the 'Related Studies' section presents problems. It's essential to include image-based methods in addition to point cloud and auxiliary data exploration for road boundaries or surfaces extraction. Following this, consider adding a new subsection reviewing image-based methods.

6.      The 'iterative weighting principle' needs proper citation if it was not proposed by the authors. Also, provide a comprehensive explanation for reader comprehension.

7.      The classification or naming rule in section 3.2.2 seems irrational. Consider simplifying the process since all points can be classified into non-curb and curb points without additional steps.

8.      Figure 1 needs improvement for better readability. Consider regenerating it with additional annotations. The interpretation of the third and fourth rows is unclear. Provide clearer explanations regarding the order and relationships between these subfigures in these two rows.

9.      Introduce the full names of "MPS," "DBSCAN," and "RANSAC" when they first appear in Figure 1 to explain their meanings.

10.   Consider revising the usage of x2-1, y2-1, and z2-1 to represent (x1,y1,z1) and (x2,y2,z2) for better clarity.

11.   Verify the annotations and content match for V12 and V21 in Figure 5.

12.   Elaborate on the first step of the DBSCAN process from line 391 to 392.

13.   The description from line 397 to 400 does not align with Figure 9(a). In the figure, points Y and Z are noted as "Core points" instead of "Border Points." Rectify this discrepancy regarding the classification of points.

14.   Please provide detailed clarification for the process mentioned from line 411 to 413.

15.   The description of road edge refinement from line 461 to 472 appears to be a repetition from the earlier description. Please revise accordingly.

16.   Consider splitting Figure 9(b) and Figure 9(c) for better clarity and center the whole figure.

17.   Please define the meaning of "n" in equation (9).

18.   Please explain the meanings of ki, ki+1, etc. and clarify whether they are assigned the same values.

19.   In Figure 11, both longitude/latitude and the north arrow convey direction redundantly. Furthermore, the relationship between the submaps and their corresponding locations isn't clearly depicted. Reconsider the use of longitude/latitude in this context for accuracy.

20.   Consider revising the use of the term 'performance' as it includes 'accuracy' evaluation in line 147.

21.   Line 377 requires a reference for RANSAC. Provide detailed information about RANSAC for readers to follow the paper.

22.   In line 68, the authors mentioned "the integration of machine learning and artificial intelligence." However, machine learning is a subset of artificial intelligence. Please cross-reference [14] and reconsider the statement.

23.   Address the incomplete sentence in line 79, ensuring it contains a complete verb phrase.

24.   Avoid using informal language like "aka" in academic writing.

Round 2

Reviewer 2 Report

Comments and Suggestions for Authors

I havenot further Comments.

Comments on the Quality of English Language

none

Author Response

Many thanks to the reviewer for their valuable contributions.

Reviewer 3 Report

Comments and Suggestions for Authors

In comparison with the initial manuscript, the authors have implemented significant modifications in the second version. However, certain areas still require refinement to improve the article's readability and conformity to standardized conventions.

1.As promising outcomes have been achieved in previous research, what is the real research gap in your work? Please reconsider this question and reorganize your abstract and introduction.

2. Regarding my prior comment 3, though the authors have moved the literature review related to activity-driven methods to an earlier section, the depth of this review remain inadequate. Furthermore, it's advisable to present this review as a separate paragraph to enhance clarity and organization.

3. Sections reviewing image-based methods (lines 133-145) are currently quite simplistic and resemble a mere enumeration of topics in publications. For instance, citations like "Yamaguchi et al. [68] developed…" lack elaboration on their results and contributions to the field. Similar issues are observed with citations [35], [37], [38], and [13]. It's crucial to provide more substantial details regarding their outcomes and contributions. Consider referencing your own citations (e.g., citing [10] and [39]) as models for thorough explanation and analysis.

4. In Figures 9(b) and 9(c) and the corresponding content in Figure 1, it's advisable to consider adding a clear boundary between the two figures for better visual distinction.

5. Regarding Figure 11, it is recommended to add a world map or incorporate the longitude and latitude indicators within the top-left sub-map to provide additional geographical context. Additionally, the use of circles as map boundaries may be perceived as informal; therefore, adopting a more standardized representation would be advisable.

6. Regarding my previous comment 22, it is important to note that deep learning is a subset of machine learning, which, in turn, falls within the broader domain of artificial intelligence (AI).

7. Building upon my previous comment 23, please ensure the sentence structure is complete. It should read, "The main contributions of this paper are as follows:".
